# Anomalously bright single-molecule upconversion electroluminescence

Yang Luo[1,5], Fan-Fang Kong [1,5], Xiao-Jun Tian[1], Yun-Jie Yu[1], Shi-Hao Jing [1], Chao Zhang[1], Gong Chen [1] ✉, Yang Zhang [1,2,3], Yao Zhang [1,2,3], Xiao-Guang Li [4], Zhen-Yu Zhang [1,2,3] & Zhen-Chao Dong [1,2,3] ✉

Efficient upconversion electroluminescence is highly desirable for a broad range of optoelectronic applications, yet to date, it has been reported only for ensemble systems, while the upconversion electroluminescence efficiency remains very low for single-molecule emitters. Here we report on the observation of anomalously bright single-molecule upconversion electroluminescence, with emission efficiencies improved by more than one order of magnitude over previous studies, and even stronger than normal-bias electroluminescence. Intuitively, the improvement is achieved via engineering the energy-level alignments at the molecule–substrate interface so as to activate an efficient spin-triplet mediated upconversion electroluminescence mechanism that only involves pure carrier injection steps. We further validate the intuitive picture with the construction of delicate electroluminescence diagrams for the excitation of single-molecule electroluminescence, allowing to readily identify the prerequisite conditions for producing efficient upconversion electroluminescence. These findings provide deep insights into the microscopic mechanism of single-molecule upconversion electroluminescence and organic electroluminescence in general.

Upconversion electroluminescence (UCEL) is an important nonlinear optoelectronic phenomenon whereby the emitted photon energy is higher than the excitation electron energy. It has been frequently observed in various ensemble electroluminescence (EL) systems[1–5] including organic light-emitting diodes[6–9]. The UCEL mechanisms there have been attributed to triplet-triplet annihilation[1,8], thermally assisted effects[5–7,9], or Auger processes[2]. Recently, UCEL in a single-molecule junction has also been demonstrated[10–13] at low temperatures by using scanning tunneling microscope-induced luminescence (STML)[12,14–32]. Since neither the thermally activated nor energy transfer mechanisms involving intermolecular interactions are likely to be operative there, this phenomenon was attributed to a spin-triplet-mediated two-electron excitation mechanism, involving an inelastic

electron–molecule scattering (IES) process to first promote the molecule to an intermediate triplet state, followed by a carrier injection (CI) mechanism to generate a singlet molecular exciton. Plasmon-assisted and vibration-assisted upconversion mechanisms can be safely ruled out as their lifetimes are too short to serve as relay states[10].

Efficient single-molecule electroluminescence that can operate at low driving voltages such as in the UCEL regime is highly desirable for nanoscale optoelectronic applications in terms of signal intensity and energy saving[33,34]. Unfortunately, its realization is challenging since the existing mechanisms that lead to efficient upconversion in ensemble systems with indispensable intermolecular coupling are not applicable to a single molecule. The single-molecule UCEL previously reported is orders of magnitude weaker than the normal-bias electroluminescence

[1]International Center for Quantum Design of Functional Materials (ICQD), Hefei National Research Center for Physical Sciences at the Microscale and CAS Center for Excellence in Quantum Information and Quantum Physics, University of Science and Technology of China, Hefei, Anhui 230026, China. [2]School of Physics and Department of Chemical Physics, University of Science and Technology of China, Hefei, Anhui 230026, China. [3]Hefei National Laboratory, University of Science and Technology of China, Hefei 230088, China. [4]Institute for Advanced Study, Shenzhen University, Shenzhen 518060, China. [5]These authors contributed equally: Yang Luo, Fan-Fang Kong. ✉e-mail: gongchen@ustc.edu.cn; zcdong@ustc.edu.cn

acquired at voltages that surpass the molecular exciton energy, due to the involvement of the inefficient IES excitation step. It therefore seems naturally plausible that the key issue for improving the single-molecule UCEL efficiency is to circumvent the inefficient IES bottleneck in the excitation sequence.

In this work, we report the first experimental realization of anomalously bright upconversion electroluminescence in a single phthalocyanine molecule through engineering the energy-level alignments within the molecular junction. By tuning the work function of the metal substrate to align the molecular energy levels properly, we have improved UCEL efficiencies by more than one order of magnitude over previous UCEL studies[10–13]. Intriguingly, the emission intensities at upconversion bias are even stronger than the normal electroluminescence excited at bias above the molecular optical gap. By analyzing differential conductance data and bias-dependent electroluminescence, we discover a new spin-triplet-mediated UCEL mechanism that only involves pure carrier injection steps, which can explain the anomalously bright UCEL without invoking the inefficient IES step. We further develop a microscopic theory based on quantum master equations to construct EL diagrams for the excitation of single-molecule electroluminescence, allowing to readily identify the prerequisite conditions for producing efficient UCEL. Our findings provide deep insights into the microscopic excitation mechanisms of organic electroluminescence, and are instructive for the design and development of energy-efficient organic optoelectronic devices.

## Results and discussion

### Anomalously bright UCEL from a single $H_2Pc$ molecule

Figure 1a illustrates schematically the STML setup, where electronically decoupled isolated free-base phthalocyanine ($H_2Pc$) molecules were excited by highly localized tunneling electrons. Figure 1b shows plasmon-enhanced fluorescence[35–38] from the lobe of a single $H_2Pc$ molecule adsorbed on the three monolayer (ML) thick NaCl surface on

Au(111) [i.e., 3ML-NaCl/Au(111)] at different voltages in opposite bias polarities (see Methods for experimental details). A bipolar molecular electroluminescence behavior is observed for this system on Au(111), which is different from the practically unipolar behavior observed on Ag(100)[10]. Two sharp emission peaks are observed at -1.81 and -1.92 eV, which can be assigned to the emission from the two lowest-lying singlet excited states $Q_x$ (or $S_1$) and $Q_y$ (or $S_2$) of the neutral $H_2Pc$ molecule, respectively[10,39].

Of particular interest is the occurrence of an over-bias UCEL phenomenon in which the molecular electroluminescence can be observed even when the energy of tunneling electrons is lower than the optical gap (or singlet exciton energy) of the molecule (i.e., $|eV_b| < E_{S_1}$) at both bias polarities. At the negative bias, the UCEL intensity at $V_b = -1.7$ V is much weaker than the normal electroluminescence intensity at $V_b = -2.0$ V, similar to our previous study for $H_2Pc$/NaCl/Ag(100)[10]. Surprisingly, at positive bias, the UCEL intensity at $V_b = 1.7$ V is much stronger than that at alike negative bias, more than two orders of magnitude higher. (Note that the UCEL intensity observed here at $V_b = 1.7$ V is also much stronger than the works reported by other groups[12,13].) Even more strikingly, the UCEL intensity is even stronger than the normal electroluminescence intensity excited at a "normal" bias (e.g., $V_b = 2.0$ V) that is defined being larger than the molecular optical gap ($E_{S_1} = 1.81$ eV). Nevertheless, by contrast, the UCEL intensity at $V_b = 1.5$ V becomes much weaker. Consequently, the abnormally strong UCEL observed at positive bias such as $V_b = 1.7$ V suggests the existence of a new UCEL channel available at this bias that can efficiently accumulate the energy of multiple electrons to excite the molecule. Note that there is a small blue shift of ~2 meV with increasing bias voltages from 1.5 V to 1.7 V, likely associated with the photonic Lamb shift (see Supplementary Fig. 1)[37,40,41].

To obtain a comprehensive picture of the molecular electroluminescence phenomenon at positive bias, we also investigate the evolution of photon emission intensities over a relatively wide range of

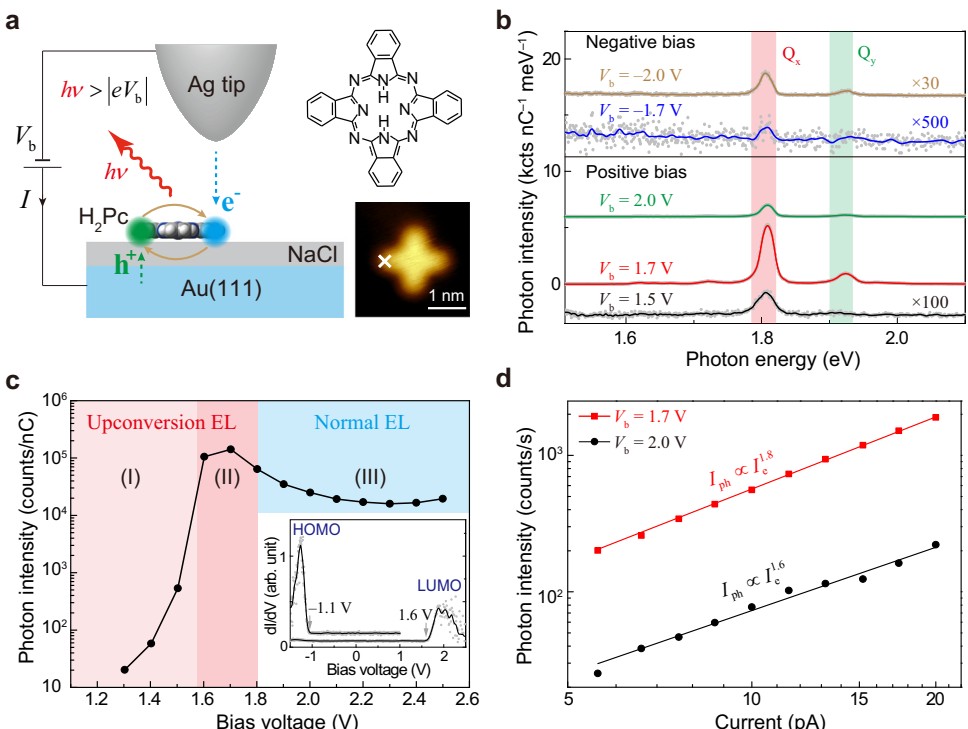

**Fig. 1 | Efficient UCEL from a single $H_2Pc$ molecule. a** Schematic of STM-induced single-molecule electroluminescence. Insets show the structure and STM image (−0.7 V, 2 pA) of $H_2Pc$. **b** STML spectra acquired from $H_2Pc$/3ML-NaCl/Au(111) at different bias ($V_b$): 1.5, 1.7, 2.0, −1.7 and −2.0 V. **c** Bias-dependent

intensity integrated over the $Q_x$ peak. Inset: Differential conductance ($dI/dV$) curve of $H_2Pc$, with the two peaks assigned to the HOMO and LUMO states. **d** Dependence of molecular emission intensities $I_{ph}$ on tunneling currents $I_e$ at $V_b = 1.7$ and 2.0 V.

bias, as depicted in Fig. 1c. The UCEL onset voltage is close to the energy of the spin-triplet state $T_1$ ($E_{T_1}$) around 1.2 eV for $H_2Pc$[42] (see Supplementary Fig. 2 for more details), suggesting the vital role of the long-lifetime $T_1$ in UCEL. Three distinct electroluminescence regions can be identified based on the $Q_x$ peak intensity as a function of applied bias voltages. Specifically, Region (I) refers to the UCEL region with low emission efficiencies for $V_b < -1.6$ V; Region (II) to the anomalously bright UCEL region for $-1.6$ V $\leq V_b < 1.81$ V; and Region (III) to the normal electroluminescence region for $V_b \geq 1.81$ V. Notably, the rapid increase of the $Q_x$ peak intensity in the upconversion region coincides with the onset voltage of the LUMO (about 1.6 V) in the differential conductance ($dI/dV$) shown in the inset of Fig. 1c, implying the carrier injection through the molecular LUMO likely playing an important role in the efficient UCEL process. A nonlinear behavior with a power exponent of ~1.8 is also obtained in the dependence of photon intensities ($I_{ph}$) on tunneling currents ($I_e$) at $V_b = 1.7$ V, which indicates the multiple-electron excitation nature in the UCEL region. Surprisingly, a similar nonlinear exponent is also observed in the $I_{ph}$–$I_e$ curve measured at 2.0 V, which implies that multi-electron excitation is also dominant even in the normal electroluminescence region.

## Mechanism for anomalously bright single-molecule UCEL

Since the anomalously bright UCEL phenomenon can be observed at very small tunneling currents (e.g., 30 pA), the higher-order tunneling mechanisms reported at very large currents[43] or even atomic contacts[44–46], involving electron–electron or electron–plasmon interactions via virtual intermediate states, appear unlikely here. This

observation, together with the detected onset voltage coinciding with the $T_1$ energy, suggests the involvement of a long-lifetime spin-triplet state[47] as the most probable option for the intermediate relay state. Figure 2 illustrates two possible triplet-mediated UCEL mechanisms (see Supplementary Notes 3 & 4 for more detailed discussions). One is the IES + CI mechanism illustrated in Fig. 2a. The molecule can be excited from the ground-state $S_0$ to the intermediate state $T_1$ via the spin-exchange IES mechanism[4,10] (see Supplementary Note 5 for more discussion), in which the tunneling electron has to exchange with the electron that possesses an opposite spin in the molecular HOMO (step 1). Then the electron in the intermediate $T_1$ state can tunnel to the substrate (step 2), leaving behind a transient cation that can be transformed to the neutral $S_1$ state upon an electron injection from the tip (step 3). Since IES is usually an inefficient excitation mechanism due to a very short electron–molecule collision time[48], the IES + CI mechanism is believed to be responsible for the low UCEL intensities observed in Region (I) when $V_b < 1.6$ V, similar to the weak UCEL observed for $H_2Pc/NaCl/Ag(100)$ at negative bias voltages[10]. However, the IES + CI mechanism cannot explain the abnormally strong UCEL observed in Region (II) ($1.6$ V $\leq V_b < 1.81$ V).

We propose here a mechanism based on multi-step pure carrier injection processes, as illustrated in Fig. 2b. In Region (II) ($1.6$ V $\leq V_b < 1.81$ V), the Fermi level of the tip lies above the molecular LUMO. In this case, the tip electron can inject into the LUMO, generating a transient anion $D_0^-$ (step 1 in the left panel of Fig. 2b). In the presence of this LUMO electron, the electron energy in the original molecular "HOMO" can shift upwards above the Fermi level of the metal substrate

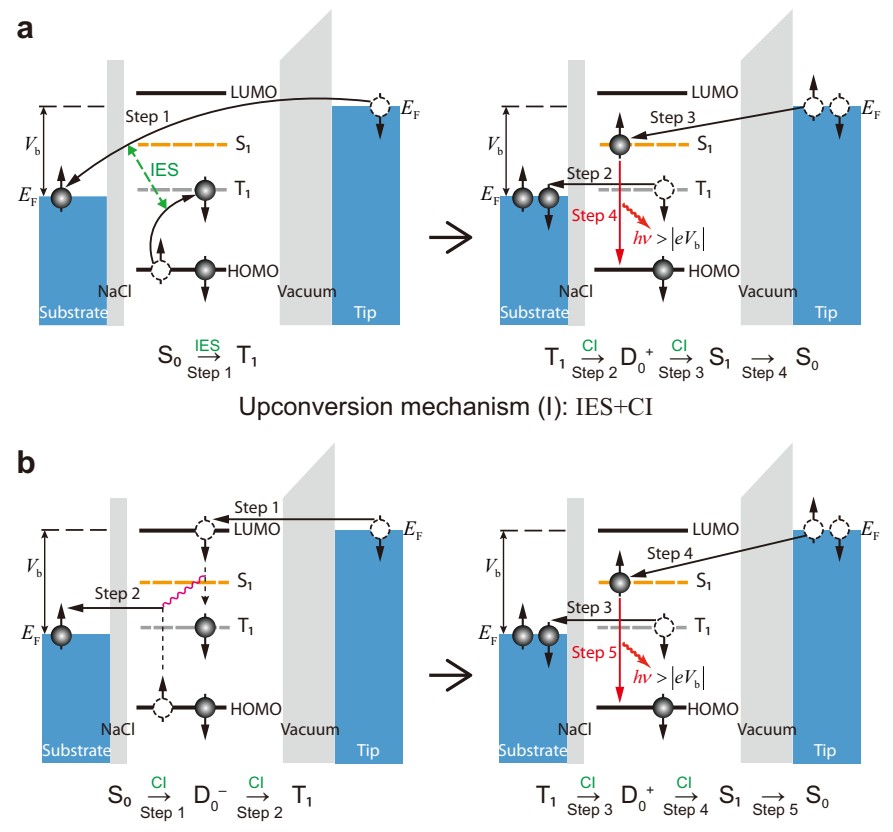

**Fig. 2 | Schematic diagrams of contrasting single-molecule UCEL mechanisms.** In the upconversion region, the first tunneling electron excites the neutral molecule from the $S_0$ ground state to the $T_1$ triplet state by either IES (**a**) or via a transient anionic state through two sequential carrier injection steps (**b**). Then, the second tunneling electron can promote the molecule from the $T_1$ state to the $S_1$ state via a transient cationic state through another two sequential carrier injection steps, as shown identically on the right panels in **a** and **b**. Here we use solid black arrows for carrier injections (CI), dashed green arrows for inelastic electron scattering (IES), red arrows for photon emission, vertical dashed lines to illustrate level shifting due to charging/discharging, and pink wavy lines to connect transitions that occur simultaneously. The same annotations are adopted for other similar figures throughout the whole manuscript. See Supplementary Notes 3 & 4 for more details.

owing to the mutual Coulombic interaction, thus allowing the "HOMO" electron to tunnel to the substrate (step 2, Supplementary Note 4). In other words, a hole in the substrate can inject into the up-shifted molecular "HOMO", leaving the molecule in the $T_1$ neutral intermediate state, but not in the $S_1$ state due to the energy conservation principle. The generation of $T_1$ requires $\phi_e \geq E_{T_1}$, where $\phi_e$ is the electron injection barrier defined by the energy difference between the molecular LUMO and the Fermi level of the substrate (Fig. 3a, see Supplementary Note 4 for more details).

Upon the generation of $T_1$, the molecule can then be excited from $T_1$ to $S_1$ through a sequential carrier injection process that involves a transient cationic state $D_0^+$ and further injection of a second electron from the tip (steps 3 & 4 in Fig. 2b, more details in Supplementary Note 3). It is evident that the long-lifetime $T_1$ state is essential for both the IES + CI and CI + CI mechanisms, but the latter is much more efficient since the whole process, including the excitation of the $T_1$ state, proceeds through pure carrier injection steps, without the involvement of inefficient IES processes. This mechanism is also believed to be dominant even in Region (III) for $V_b \geq 1.81$ V, because no sharp change in the electroluminescence intensities happens at around 1.81 V. Indeed, the nonlinear relationship between the electroluminescence photon intensity $I_{ph}$ and the tunneling current $I_e$ at $V_b = 2.0$ V does support the validity of the same two-electron CI + CI mechanism. Here the decrease of the electroluminescence intensity with increasing voltages is probably due to the increased tip–molecule distance, which decreases the electroluminescence efficiency due to the reduced plasmon–exciton coupling strength[37,49]. Nevertheless, the CI + CI upconversion electroluminescence mechanism proposed here is essentially correct since the UCEL intensity jumps by over two orders of magnitude as the excitation mechanism changes from IES + CI to CI + CI at -1.6 V (Fig. 1c) while the gap distance varies only slightly (see Supplementary Note 6 for details). We would like to note that the triplet–charge interaction mechanism[50,51] involving intermolecular charge transfer was used to explain the upconversion process from $T_1$

to $S_1$ in organic light-emitting diodes[52], but is unlikely to account for the single-molecule UCEL phenomenon here (see Supplementary Note 3 for more details).

## General microscopic model for single-molecule electroluminescence

The CI + CI mechanism proposed in Fig. 2b qualitatively explains the anomalously bright UCEL in $H_2Pc/NaCl/Au(111)$. The model can also account for the absence of UCEL for $H_2Pc/NaCl/Ag(100)$ at positive bias because the criterion $\phi_e \geq E_{T_1}$ is no longer satisfied when the substrate is changed from Au(111) to Ag(100) with a smaller work function. Such a sharp contrast suggests that the energy-level alignment at the molecule–substrate interface (Fig. 3a) is likely to play a critical role in the organic electroluminescence behavior. Therefore, it is highly desirable to construct a comprehensive theory on the correlation between the energy-level alignment and the excitation mechanisms so that one can have a panoramic view on how to generate energy-efficient electroluminescence, including anomalously bright UCEL. To this end, we propose a microscopic model based on the quantum master equation to describe the excitation and decay dynamics of a single molecule in a biased tunneling junction, which enables to construct EL diagrams for the excitation behavior of single-molecule electroluminescence (see Supplementary Note 5 for details).

The central physical quantity in our model is the excitation efficiency $\eta_{ex}$ for converting the tunneling electron energy to molecular excitons. Figure 3b showcases an EL diagram with $\eta_{ex}$ numerically evaluated as a function of $V_b$ and $\phi_e$, assuming $E_{T_1} = 1.2$ eV, $E_{S_1} = 1.8$ eV, and $\phi_h = 1.1$ eV so as to cover the energy-level alignment configuration for $H_2Pc/3ML$-NaCl/Au. The EL diagram can be classified into distinctly different areas where $\eta_{ex}$ differs by many orders of magnitude, thus clearly revealing the close relationship between the exciton excitation behavior of a molecule and its energy-level alignment. Three regimes can be categorized in the EL diagram in terms of bias voltages: no electroluminescence when $|eV_b| < E_{T_1}$, UCEL when $E_{T_1} \leq |eV_b| < E_{S_1}$, and normal electroluminescence when $|eV_b| \geq E_{S_1}$. The normal electroluminescence is usually dominated by the one-electron excitation mechanism while the UCEL is accomplished by the spin-triplet-mediated two-electron excitation processes.

Let us first take a look at typical one-electron excitation mechanisms illustrated in Fig. 3c. Once $|eV_b| \geq E_{S_1}$ and no molecular orbitals are located inside the bias window, the molecule can be excited to $S_1$ via the inefficient IES mechanism[4,10]. When electron tunneling through molecular orbitals becomes feasible ($eV_b \geq \phi_e$ or $eV_b \geq -\phi_h$), the IES excitation channel will be suppressed because the proportion of the branching current that tunnels directly between the tip and substrate is significantly reduced. This mechanism is termed as weak-IES (or $w$-IES). The most efficient approach to excite the molecule to $S_1$ (i.e., to generate the strongest electroluminescence) is the CI mechanism, in which a molecule is excited through two sequential CI steps via a transient charged state[10,11,23,25,32,53]. This CI mechanism becomes possible at positive bias via a transient anion when $eV_b \geq \phi_e \geq E_{S_1}$; or at negative bias via a transient cation when $|eV_b| \geq \phi_h \geq E_{S_1}$ (see Supplementary Notes 5 & 7 for more details).

By contrast, the excitation mechanisms in the UCEL regime ($E_{T_1} \leq |eV_b| < E_{S_1}$) are much more complicated, because the excitation to $S_1$ involves spin-triplet-mediated two-electron processes (namely, from $S_0$ to $T_1$ and then from $T_1$ to $S_1$) and in principle, any of the three one-electron excitation mechanisms described in Fig. 3c is possible for each excitation step. In this regard, the theoretically simulated EL diagram allows us to directly distinguish various UCEL regions in the EL diagram and identify corresponding excitation mechanisms. As shown in Fig. 3b, four different UCEL mechanisms can be identified: $w$-IES+$w$-IES, $w$-IES + CI, IES + CI and CI + CI. Most of these excitation processes involve one or two IES (or $w$-IES) steps, resulting in low $\eta_{ex}$. Only in the special case where $E_{S_1} > \phi_e \geq E_{T_1}$ and $eV_b \geq \phi_e$ (red triangular area in

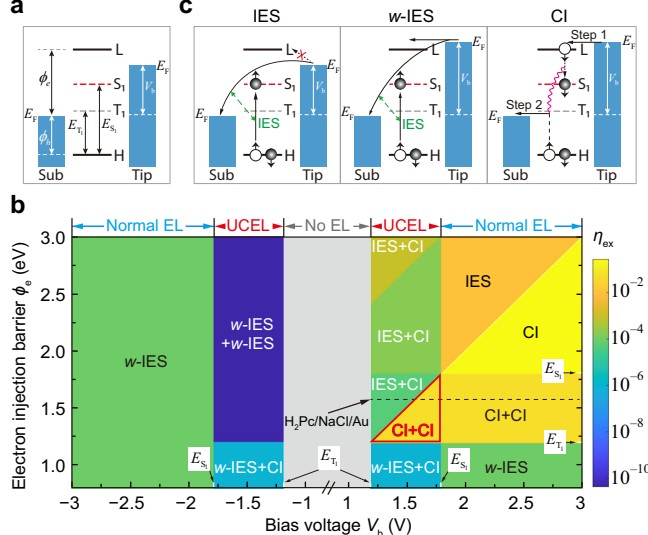

**Fig. 3 | Simulated EL diagrams for single-molecule electroluminescence.**
**a** Energy-level diagram for a single-molecule junction. $E_{T_1}$ ($E_{S_1}$) is the energy of the lowest excited spin-triplet (spin-singlet) states, $\phi_e$ ($\phi_h$) is the electron (hole) injection barrier defined as the energy difference between the molecular LUMO (HOMO) and the Fermi level of the substrate. The molecular levels are assumed to be pinned to the substrate. **b** Numerically simulated exciton excitation efficiency $\eta_{ex}$ as a function of $V_b$ and $\phi_e$ based on the quantum master equation model. In each region, only the dominant mechanism is highlighted. The simulation parameters are given in Supplementary Note 5. **c** Schematics for different one-electron excitation mechanisms.

Fig. 3b), the CI + CI mechanism involving pure carrier injection steps becomes possible and is responsible for the anomalously bright single-molecule UCEL.

Specifically, regarding the electroluminescence behavior of the H$_2$Pc/3ML-NaCl/Au(111) system at positive bias (indicated by the horizontal dashed line in Fig. 3b), one can see that the electroluminescence mechanism changes from IES + CI to CI + CI when $eV_b = \phi_e$ at ~1.6 eV, which explains the experimentally observed sharp increases in the electroluminescence intensity at ~1.6 V. However, even in the normal electroluminescence region ($V_b \geq 1.81$ V), only the low-efficiency $w$-IES mechanism is available for the one-electron excitation process, while the CI mechanism is energetically inhibited. Thus the CI + CI mechanism is still dominant in this region. It should be noted that by assuming $\phi_h \leq E_{T_1}$, Fig. 3b only shows a special case of the complete EL diagram. In addition, to understand the electroluminescence phenomenon at negative voltages for H$_2$Pc/3ML-NaCl/Au(111), the excited cationic state should also be considered (see Supplementary Note 7.1.2 for details). We would like to note that different STML phenomena were reported for the "same" H$_2$Pc/3ML-NaCl/Au(111) system by Rai et al[13]., probably due to different adsorption configurations of H$_2$Pc molecules on NaCl and resultant different energy-level alignments at the molecular interface (see Supplementary Note 7.1.3 for details).

## Tuning of driving voltages for UCEL

With the EL diagrams thus constructed, the mechanisms and prerequisite conditions for producing UCEL can be easily identified. In addition, the EL diagrams can also be used to predict the electroluminescence behavior for a given material system as long as the molecular exciton energy and differential conductance data are available, as exemplified above for the anomalously bright UCEL of the H$_2$Pc/3ML-NaCl/Au(111) system. Furthermore, the CI + CI area marked by the red triangle in Fig. 3b clearly indicates that the driving voltage for generating anomalously bright UCEL can be tuned through engineering energy-level alignments, as long as $E_{S_1} > \phi_e \geq E_{T_1}$ and $eV_b \geq \phi_e$.

Figure 4 showcases two examples of fine tuning the electron injection barrier $\phi_e$ through the simple variation in the NaCl spacer thickness because such thickness affects the image potential effect on the molecule produced by the metal substrate[54–56]. As shown in the upper panels of Figs. 4a and 4b for H$_2$Pc/NaCl/Au(111), accompanying the spacer thickness change from 3 ML to 2 ML, the onset voltage of LUMO (i.e., $\phi_e$) decreases from ~1.6 to ~1.5 V, leading to the downward shifting from ~1.7 to ~1.6 V for the driving voltage at which the UCEL intensity is maximized. (As a side note, the negative differential conductance is observed at higher bias.) Similarly, For PtPc/NaCl/Au(111), when the spacer layer is changed from 4 ML to 3 ML, the LUMO onset voltage decreases from ~1.8 to ~1.65 V, yielding a downward shifting in the optimal driving voltage from ~1.9 to ~1.8 V. Note that these excitation voltages are still in the UCEL region since PtPc has a singlet exciton energy at ~1.95 eV (see Supplementary Fig. 13). Figure 4c illustrates schematically the fine tuning of the energy-level alignment through the above control of spacer layer thickness, which can be realized through the design of molecular electronic structures and metal substrates, or both. The observed parallel evolution regarding the LUMO positions and optimal UCEL driving voltages gives further justification for the validity of our theoretical model.

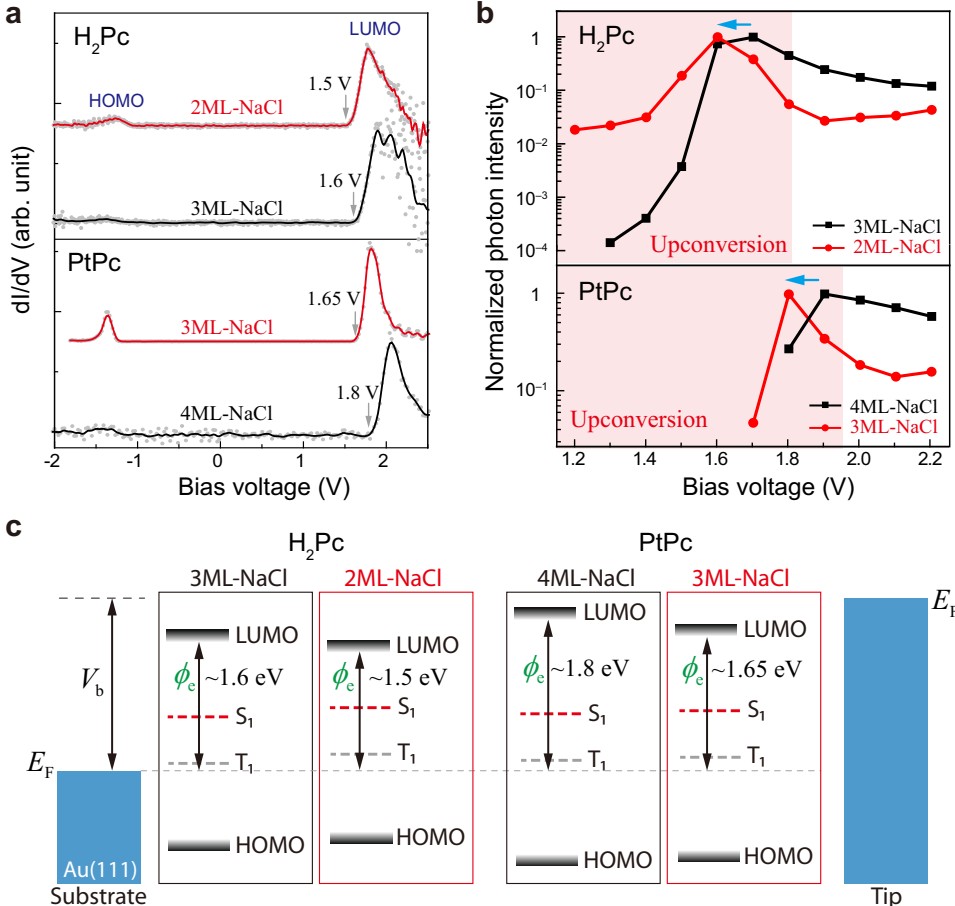

**Fig. 4 | Tuning driving voltages for anomalously bright UCEL.** Differential conductance (d$I$/d$V$) (**a**) and bias-dependent electroluminescence intensities integrated over the S$_1$ peak (**b**) for H$_2$Pc (upper panel) and platinum phthalocyanine (PtPc, lower panel) molecules with different NaCl thicknesses on Au(111). **c** Corresponding energy-level diagrams.

In summary, we have experimentally demonstrated anomalously bright single-molecule UCEL via controlled engineering of energy-level alignments at the interfaces and explained it by proposing an intriguing spin-triplet-mediated UCEL mechanism that involves efficient excitation via pure carrier injection steps. Based on the quantum master equations, we have further developed a delicate microscopic model to construct EL diagrams for the excitation of single-molecule electroluminescence, which provide a vivid and straightforward view on the relationship between the energy-level alignment of a biased single-molecule junction and its diverse electroluminescence behaviors. For the anomalously bright UCEL to occur, the excitations from $S_0$ to $T_1$ and from $T_1$ to $S_1$ via the CI + CI mechanism require critical energy-level alignments at the interfaces, which enables the amazing transitions between different charge states, namely, neutral ground-state singlet $S_0 \rightarrow$ anionic $D_0^- \rightarrow$ triplet $T_1 \rightarrow$ cationic $D_0^+ \rightarrow$ neutral excited-state singlet $S_1$. This is probably why such anomalously bright single-molecule UCEL has not been reported before and the underlying microscopic mechanism is so difficult to discover. These findings offer indispensable microscopic insights into the electro-optic conversion processes at the single-molecule level, and should be instructive in optimizing single-molecule-based optoelectronic devices and organic electronics beyond the present systems.

## Methods
In the present study, the experiments were conducted in a custom optical STM (Unisoku) operated at low temperatures (-7 K) and ultrahigh vacuum conditions (< $10^{-10}$ Torr). The Au(111) and Ag(100) substrates were cleaned by cycles of sputtering and annealing. Ultrathin insulating NaCl films were grown on the metal substrate through a home-built evaporator, producing (100)-terminated NaCl islands with a thickness of two-to-four monolayers (ML) that serve as a spacer layer to prevent the quenching of molecular fluorescence by the metal substrate. Isolated $H_2Pc$ (or PtPc) molecules were then deposited onto the NaCl islands at ~7 K through in-situ evaporation from a Knudsen cell. (The selection of planar phthalocyanine molecules for this research was based on several compelling factors. Both the free-base and metal phthalocyanine molecules are widely utilized in the STML community due to their well-defined geometry for identification by STM imaging, remarkable stability for adsorption on NaCl spacer layers, rigid structure and resultant comparatively high electroluminescence intensity.) The silver tips used for STM imaging and STML experiments were prepared by electrochemical etching and cleaned by electron bombarding and argon-ion sputtering[57,58]. Molecular electroluminescence was collected with a built-in lens close to the STM junction and analyzed with a liquid-nitrogen-cooled charge-coupled device spectrometer (Princeton Instruments)[10,19]. All the STM imaging and STML spectra were measured at the constant-current mode. All spectra presented in this work were not corrected for the wavelength-dependent sensitivity of photon detection systems. $dI/dV$ spectra were obtained using the lock-in technique, with a bias modulation of 20 mV at 329 Hz.

## Data availability
All study data are included in the main text and the supporting information. Source data are provided as a Source Data file. Source data are provided with this paper.

## Code availability
Source code and scripts performed for the study have now been uploaded to: https://github.com/ustcamp/EL_diagram.git

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

## Acknowledgements

This work was supported by the National Natural Science Foundation of China (Grant Nos. 12334018, 21790352, 22174135, 11874268, 12374033, 12004343, 11974323), the National Key R&D Program of China (Grant Nos. 2016YFA0200600, 2021YFA1500500), the Strategic Priority Research Program of Chinese Academy of Sciences (Grant No. XDB36000000), Anhui Initiative in Quantum Information Technologies (Grant No. AHY090000), and Innovation Program for Quantum Science and Technology (Grant Nos. 2021ZD0303301, 2021ZD0302800).

## Author contributions

Z.C.D. conceived and supervised the project. Y.L., F.F.K., X.J.T., Y.J.Y. and S.H.J. performed experiments and analyzed data. Y.L. and G.C. derived the theory and performed theoretical simulations. Y.L., G.C., C.Z., Yang Zhang, Yao Zhang, X.G.L., Z.Y.Z., and Z.C.D. contributed to the data interpretation. Z.C.D., Z.Y.Z., X.G.L., Y.L. and G.C. co-wrote the manuscript. All authors discussed the results and commented on the manuscript.

## Competing interests

The authors declare no competing interests.
