## [Peer Review File · Nature Communications]

Anomalously Bright Single-Molecule Upconversion ElectroluminescenceREVIEWER COMMENTS

Reviewer #1 (Remarks to the Author):

This manuscript presents a study focusing on anomalously bright single-molecule upconversion electroluminescence by STM. Based on bias-dependent experiments (including dI/dV to determine the electron injection to LUMO and plot of emission intensities versus current to determine the two-electron involved process) and control experiments with different molecules and different-width NaCl layers, as well as the theoretical quantum master equations, the authors identified a pure charge injection mechanism, which enables a high upconversion efficiency at $ET1 < |eVb| < ES1$. The experiments and data were quite impressive and I really appreciate the detailed construction of the electroluminescence diagram at different ranges of bias voltage and charge injection barrier. I still have some questions that need to be clarified. I would suggest the publication of this paper after the following issues are addressed.

The photons with ~ 1.81 eV (Qx peak) were obtained at bias voltage-dependent measurements (Fig. 1b). However, the Stark effect on the molecular orbital by the high electric field (nanogaps with $1.5\sim 2$ V bias voltage) cannot be ignored (such as Science 2021, 373, 95–98), especially at the low temperature (7 K). Here, the energy level was determined, including $ET1 = \sim 1.2$ eV, $ES1 = \sim 1.8$ eV, $ECoul = \sim 0.9$ eV, and $Eex = \sim 0.6$ eV. I wonder if the regulation of these molecular orbitals by the electric field is considered in the theoretical simulations. In addition, a blue shift of the emitting photons was observed with increasing bias voltage from 1.5 V to 1.7 V (Fig. 2b). What is the reason for this?

Long-lived triplet state, as an intermediate relay state, enables upconversion emission based on multi-electron injection. Is it possible to detect the emission of phosphorescence ($T1$ to $S0$) experimentally? Because the direct relaxation from $T1$ to $S0$ is spin-forbidden, the phosphorescence of the H2Pc may be weak. However, the heavy atom Pt in PtPc would provide a spin-orbital coupling and allow the phosphorescent emission. In addition, the excitation from $S0$ to the intermediate state $T1$ via the spin-exchange IES mechanism also appears to be spin-forbidden, especially for the H2Pc without heavy atom. The detailed mechanism of the direct excitation of spin triplets should further be discussed.

Based on the previous question, I wonder that if the key intermediate relay state (such as the charged state and the $T1$ state) as well as the alignment between molecular orbital and Fermi energy level of electrodes could be detected or characterized experimentally? I admit that the characterization to the key intermediate in the multi-step electron transport is challenging, but would provide more evidences to support the electroluminescence diagram.

Considering the Ag tip and metal substrate at the experiment, how does the author exclude plasmon-assisted multi-step inelastic scattering scheme (Nat. Photon. 2010, 4, 50–54)? The higher vibrational states may be populated and then enable the upconversion electroluminescence.

In the SI, with populated LUMO, the original molecular HOMO can shift upwards owing to the mutual Coulomb interaction. Therein the ΔE_{HOMO} was calculated as $E_{\text{Coul}} + E_{\text{ex}} = \sim 1.5$ eV. This a relatively large shift of the molecular orbital (refer to J. Am. Chem. Soc. 2018, 140, 70–73, ~ 0.2 eV of HOMO shift by Coulomb interaction). The detailed mechanism should further be discussed.

In Fig. S6, the top panel of IE+CL (via D0⁻), Step 2 shows the electron transport from tip to HOMO. Considering the theory of Marcus electron transfer, the larger energy difference between the electron donor and acceptor would cause a relatively slow rate of electron transfer. In other word, the energy level mismatch to some extent (although it does not violate the energy conservation). Is this electroluminescence diagram more efficient than multi-step IES?

A minor suggestion is the caption of the arrows. There are many arrows in Figures, including electron transfer (solid black), the energy transfer (pink), the excitation (dash black) and photon emission (red), IES (green).... The corresponding illustration of these arrows should be provided in the captions.

Reviewer #2 (Remarks to the Author):

Report on “Anomalously Bright Single-Molecule Upconversion Electroluminescence” by Yang Luo et al.

The authors report on an anomalously strong upconversion photon intensities for H2Pc molecules on 2 and 3 ML NaCl/Au(111). The paper is well written and the proposed model for the mechanism of upconversion is sound. The authors speak of a two orders of magnitude stronger upconversion efficiency than in previous studies which is even stronger than electroluminescence in the normal bias region.

I cannot recommend publication in the present form, as many of the claims are not substantiated by numbers and only a qualitative discussion is given. Many obvious questions remain unanswered, published literature is not considered and probably mayor points are hidden is some small remarks. I list my comments and questions below in more detail. Nevertheless, if the authors can resolve these issues, the paper might be accepted in Nature Communication.

1. Novelty and correctness of mechanism

The mechanism of upconversion via triplet formation is not new and has been discussed in the literature (see PRL 122, 177401 by the group of the author). In this respect, the manuscript has nothing new to offer. Claims of novelty in the manuscript regarding the mechanism are not justified (page 7 line 158ff). This takes away one of the mayor claims of the paper.

The news is essentially related to the high efficiency of the upconversion. This requires quantitative comparison (see below). A purely qualitative comparison with the emission in the normal region can result in a relatively bright upconversion, if the emission in the normal region is abnormally low. When the bias exceeds the photon energy, the two step mechanism should not be stopped. It should be operative for all high bias voltages, as the tunneling electrons have a broad energy distribution between the two Fermi levels of the electrodes. The presented model does not account for a drop of light emission at higher biases and the question why the intensity decreases for high bias is not explained in the manuscript. However, an interesting side note (negative differential conductivity on page 13) may explain this. It seems that at high bias, the transport channel is cut (total current drops with bias voltage, i.e. negative differential conductance) and with it the light emission, e.g. by simply charging of the molecule. This could explain a very inefficient emission under normal condition that could be mistaken as a very efficient upconversion.

2. Quantitative data

The authors do not provide quantitative data on the phonon spectra. In most of the figures, they show spectra as function of photon energy in units of counts/nC. Correct units for spectral data would be e.g. counts/(charge*photon energy). A spectrum reflects the photon spectral density as function of photon energy. Only the integral over the photon energy has the dimensions of counts/nC. Depending on the bin size of the spectra, the plotted numbers can correspond to high or low photon flux. The strong claim of the authors call for a comparison of their intensities quantitatively with results of other systems and other groups in meaningful physical units. Without this direct comparison, the claim of unusually high efficiencies of upconversion cannot be held. I also urge the authors not to normalize the photon intensities but to give the real data (Fig. 1c, Fig. 4b etc.).

3. Comparison with published data of the very same system

It remains totally unclear, why the authors do not compare their data with already published results on the very same system of H2Pc/NaCl/Au(111) (Nano Lett. 20, 7600 (2020)). This work also includes results of upconversion with spectral densities of about 400 counts/(nC meV) and in the "normal" emission of about 2000 counts/(nC meV), i.e. much higher than the upconversion. Obviously, this report is at odds

with this work and this discrepancy needs a detailed discussion. It remains completely unclear, why the two works show such vastly different efficiencies although they are on the very same material system.

4. Tip condition and negative differential conductance

The $z(V)$ curve in the supplementary figure S4 looks strange. It seems that the tip retracts by about 500pm when ramping the voltage. This is a very large change if keeping in mind, that 100pm change of tip-sample distance changes the current by about one order of magnitude in STM with materials of typical work functions. Can the authors exclude that the tip is partly insulating or it crashes into the sample at low bias? The z -position strongly reacts on the bias in the region of interest between 1.6 and 1.8 V. This corresponds to the onset of the LUMO which in turn can drive the triplet formation (mechanism II), but up to 2.4 V the z -position only increases. The mentioned negative differential conduction must lead to a decrease of current with bias (this is the definition of negative differential conductance) and the the z -position under feedback needs to approach. Why is this not observed? The two measurements in this manuscript contradict each other.

5. Comparison with negative bias voltage

The authors also supply data on the photon emission in negative bias voltage. A very bright Qx line is reported, but Figure S7 does not give quantitative information on the intensities. How do the intensities compare between positive and negative bias, quantitatively? Does this comparison support that upconversion is particularly efficient at positive bias voltages or does it suggest a particularly low efficiency in the normal region? How is the upconversion explained at negative bias voltages? Is that in agreement with LUMO transitions as well? If not, how sure can we be that the LUMO involvement at positive bias is correct when upconversion is also seen at negative bias?

6. Comparison to NaCl/Ag(100)

The data shown in the supplementary material (S9) for the analogous experiments on Ag(100) has many orders of magnitude more efficient light emission in the normal region (S9a shows Qx intensities that are indeed high and of the order of 10^8 counts/nC). Also this speaks for a strongly inefficient emission in the normal bias regime for the case of Au(111) and against a very efficient upconversion.

7. Citations

Upconversion has been studied by many groups. I find that the citations are too much biased regarding the authors own publications. Also the claim of anomalous upconversion efficiency needs to be tested against the literature especially of competitors and not exclusively to own work.

Reviewer #3 (Remarks to the Author):

In this work, efficient upconversion electroluminescence (UCEL) was observed in a single phthalocyanine molecule, and the emission efficiencies can be improved via engineering the energy-level alignments at the molecule–substrate interface. Moreover, a new spin-triplet mediated UCEL mechanism was proposed for producing efficient UCEL. These findings provide deep insights into the microscopic mechanism of single-molecule UCEL and organic electroluminescence in general.

Overall, the research is interesting and should be accepted for publication in Nature Communication after addressing several questions related.

1. Why was the phthalocyanine molecule selected for this research? 2. Does the PtPc show more efficient UCEL efficiency?

Point-by-Point Response to Reviewers' Reports on Nature Communications manuscript NCOMMS-23-38747

We thank all reviewers for their highly expertized reviews of this manuscript, which have allowed us to improve the presentation of this work in several substantial aspects. We are also grateful to all reviewers for their valuable comments and constructive suggestions. We are pleased that both Reviewers 1 and 3 are highly positive about our work and recommend its acceptance after addressing their questions and comments. Reviewer 2 raised more substantive issues to be addressed, but overall is also inclined to support acceptance of this work by stating that “*if the authors can resolve these issues, the paper might be accepted in Nature Communication*”.

As detailed below, we provide point-by-point responses to their constructive comments and suggestions, and have incorporated those points into the revised manuscript and Supplementary Information (for better clarity, the revised contents are marked in blue in the manuscript, but in bold in the response letter). With these improvements, we hope the paper is now acceptable for publication in *Nature Communications*.

Responses to Reviewer 1:

General Comment: *This manuscript presents a study focusing on anomalously bright single-molecule upconversion electroluminescence by STM. Based on bias-dependent experiments (including dI/dV to determine the electron injection to LUMO and plot of emission intensities versus current to determine the two-electron involved process) and control experiments with different molecules and different-width NaCl layers, as well as the theoretical quantum master equations, the authors identified a pure charge injection mechanism, which enables a high upconversion efficiency at $ET1 < |eVb| < ES1$. The experiments and data were quite impressive and I really appreciate the detailed construction of the electroluminescence diagram at different ranges of bias voltage and charge injection barrier. I still have some questions that need to be clarified. I would suggest the publication of this paper after the following issues are addressed.*

Response: We thank the reviewer very much for his/her appreciation of our work, positive recommendations, as well as highly constructive suggestions. In the following, we provide a point-by-point response to his/her comments.

Comment 1: *The photons with ~ 1.81 eV (Q_x peak) were obtained at bias voltage-dependent measurements (Fig. 1b). However, the Stark effect on the molecular orbital by the high electric field (nanogaps with 1.5~2 V bias voltage) cannot be ignored (such as Science 2021, 373, 95–98), especially at the low temperature (7 K). Here, the energy level was determined, including $ET1 = \sim 1.2$ eV, $ES1 = \sim 1.8$ eV, $ECoul = \sim 0.9$ eV, and*

E_{ex} = ~ 0.6 eV. I wonder if the regulation of these molecular orbitals by the electric field is considered in the theoretical simulations. In addition, a blue shift of the emitting photons was observed with increasing bias voltage from 1.5 V to 1.7 V (Fig. 1b). What is the reason for this?

Response: We thank the reviewer’s insightful comment regarding the observed Q_x peak energy and blue shifts in the STML spectra with increasing bias voltages from 1.5 V to 1.7 V.

Indeed, the peak position in STML spectra could be affected by several factors, including the Stark effect, the photonic Lamb shift [*Phys. Rev. X* 12, 011012 (2022); *Science* 373, 95 (2021); *Nat. Photon.* 14, 693 (2020)], and in the special case of H₂Pc/NaCl/Au(111), the splitting of the Q_x band for two differently orientated H₂Pc tautomers with respect to linear Moiré patterns resulting from the incommensurate NaCl and Au(111) lattices [*Nat. Nanotechnol.* 15, 207 (2020)]. The Stark effect on the peak shift from 1.5 V to 1.7 V is believed to be negligible in the present system, as rationalized below. According to the paper by Imada *et al.* [*Science* 373, 95 (2021)], the Stark effect was observed to cause a redshift of only ~1 meV when changing the voltage over a much wider range from -3 V to +1 V. In main-text Fig. 1(b) in the previous version of the manuscript, there was an 8-meV blue shift in the Q_x emission peak in the STML spectrum when the bias voltage was increased from 1.5 V to 1.7 V. This process is accompanied by an increase in the gap distance of ~0.15 nm to keep the tunneling current constant [see Supplementary Fig. S7(a)]. Thus, the increases in the bias voltage and the gap distance have opposite effects on the electric field and are believed to largely cancel out with each other, leading to similar dc electric field strengths. Therefore, in our present work, the peak shift caused by the Stark effect is believed to be very minor, much smaller than 1 meV when the bias voltage is increased from 1.5 V to 1.7 V. Since such a small Stark shift has a negligible influence on the energy levels of molecular orbitals including HOMO and LUMO, the regulation of these molecular orbitals and related state energies by the electric field is not considered in our theoretical simulations.

Fig. R1. STML spectra acquired from H₂Pc/3ML-NaCl/Au(111) above the molecular lobe at three different experimental conditions. Black curve: 1.5 V, 10 pA, 60 s, blue curve: 1.5 V, 10

pA, 600 s, red curve: 1.7 V, 30 pA, 60 s. Spectral curves are offset for clarity.

Having excluded the possibility of the Stark effect being responsible for the 8-meV blue shift shown in main-text Fig. 1(b) in the previous version of the manuscript, such a large blue shift is also found not to originate from the photonics Lamb shift. In addition to the dataset shown in main-text Fig. 1(b) in the previous version of the manuscript, we also carried out several other similar measurements at 1.5 V but with different exposure time. All others show a smaller blue shift of ~ 2 meV, as exemplified in Fig. R1. Such a large variation in the blue shift is unlikely to come from photonic Lamb shifts for a tip positioned at a given molecular position with similar changes in the tip height. In other words, the photonic Lamb shift may be responsible for the consistent blue shift of ~ 2 meV, while the abnormally large blue shift of ~ 8 meV is likely due to the splitting of the Q_x band for two differently orientated H_2Pc tautomers with respect to linear Moiré patterns resulting from the incommensurate NaCl and Au(111) lattices [*Nat. Nanotechnol.* 15, 207 (2020)]. Such a phenomenon occurs when the tip is positioned at slightly different positions or when the tip positions are shifted during the STML measurements. We thank the reviewer again for pointing out this insightful issue on the peak shift.

In the revised manuscript, in order to avoid the misunderstanding caused by the abnormally large peak shift due to possibly different tip positions, we have replaced the STML spectrum at 1.5 V (black curve in previous Fig. 1(b)) with a typical spectrum collected at the same voltage. We also added a sentence in the main text to note the peak shift issue (page 5, lines 113–115): “**Note that there is a small blue shift of ~ 2 meV with increasing bias voltages from 1.5 V to 1.7 V, likely associated with the photonic Lamb shift (see Supplementary Section S1)^{37,40,41}”, with a broadened discussion in the newly-added Supplementary Section S1. A few sentences are also added there (also in Supplementary Section S5) to note that the electric field effect is not considered in the theoretical simulations. Nevertheless, we would like to point out that this peak shift issue does not affect the main findings (including mechanisms) presented in this work since the crucial data to support these conclusions are the peak intensities at different bias voltages, which behave essentially the same as the previous dataset.**

Comment 2: *Long-lived triplet state, as an intermediate relay state, enables upconversion emission based on multi-electron injection. Is it possible to detect the emission of phosphorescence ($T1$ to $S0$) experimentally? Because the direct relaxation from $T1$ to $S0$ is spin-forbidden, the phosphorescence of the H_2Pc may be weak. However, the heavy atom Pt in $PtPc$ would provide a spin-orbital coupling and allow the phosphorescent emission. In addition, the excitation from $S0$ to the intermediate state $T1$ via the spin-exchange IES mechanism also appears to be spin-forbidden, especially for the H_2Pc without heavy atom. The detailed mechanism of the direct excitation of spin triplets should further be discussed.*

Response: We greatly appreciate the reviewer’s insightful comment regarding the detection of phosphorescence associated with the long-lived intermediate triplet state as well as the excitation mechanisms involved. We have carefully examined all collected emission spectra of the PtPc/NaCl/Au(111) system, and no discernable phosphorescence peaks were found. Nevertheless, we did detect a very weak phosphorescence peak at ~ 1.27 eV in the case of PtPc/NaCl/Ag(100), as depicted in Fig. R2. (Note that the relative intensity of phosphorescence with respect to the fluorescence might be affected by the resonance condition of the nanocavity plasmon mode.) The absence of discernible phosphorescence from PtPc on NaCl/Au(111) may be attributed to the larger optical losses associated with gold compared with silver, resulting in weaker photon emission intensities. On the other hand, no discernable phosphorescence peaks were found in the STML spectra for H₂Pc on both NaCl/Au(111) and NaCl/Ag(100) substrates, due to the very weak spin-orbit coupling.

Fig. R2. STML spectra from PtPc/4ML-NaCl/Ag(100). Experimental conditions: -2.8 V, 30 pA, 30 s.

In the following, we explain qualitatively why the direct excitation from S_0 to T_1 via the spin-exchange IES mechanism is not spin-forbidden. Since the IES induced transition rates involve complex many-body interactions and their accurate evaluation can be quite demanding, here we only try to rationalize its occurrence using simple arguments. Fig. R3 schematically illustrates the spin-exchange inelastic scattering process between an incident electron and a single molecule where the molecule is excited from the singlet ground state to the spin-triplet state. For simplicity, only the two electrons in the HOMO are explicitly considered. The wavefunction for the initial state, consisting of the incident free electron and the ground-state molecule, can be written as

$$\begin{aligned}\Psi_i(r_1, r_2, r_3) &= \phi_{k_i}(r_3)\chi_{\uparrow}(3) \times \psi_{S_0}(r_1, r_2) \\ &= \phi_{k_i}(r_3)\chi_{\uparrow}(3) \times \varphi_H(r_1)\varphi_H(r_2) \frac{1}{\sqrt{2}} [\chi_{\uparrow}(1)\chi_{\downarrow}(2) - \chi_{\downarrow}(1)\chi_{\uparrow}(2)]\end{aligned}$$

Here $\varphi_{\text{H}}(r_1)\varphi_{\text{H}}(r_2)$ and $\phi_{k_i}(r_3)$ are the spatial part of the wavefunctions for the S₀-state molecule and the incident electron (with k_i denoting the momentum), respectively, $\chi_{\uparrow/\downarrow}$ is the wavefunction for spin. The final state consists of the scattered free electron and the molecule in the triplet state, with its wavefunction written as

$$\begin{aligned}\Psi_f(r_1, r_2, r_3) &= \phi_{k_f}(r_2)\chi_{\downarrow}(2) \times \psi_{\text{T}_1}(r_1, r_3) \\ &= \phi_{k_f}(r_2)\chi_{\downarrow}(2) \times \frac{1}{\sqrt{2}}[\varphi_{\text{H}}(r_1)\varphi_{\text{L}}(r_3) - \varphi_{\text{L}}(r_1)\varphi_{\text{H}}(r_3)]\chi_{\uparrow}(1)\chi_{\uparrow}(3).\end{aligned}$$

Here $\frac{1}{\sqrt{2}}[\varphi_{\text{H}}(r_1)\varphi_{\text{L}}(r_3) - \varphi_{\text{L}}(r_1)\varphi_{\text{H}}(r_3)]$ and $\phi_{k_f}(r_2)$ are the spatial part of the

wavefunctions for the T₁-state molecule and the scattered electron (with k_f denoting the momentum), respectively. Note that to simplify our discussion, here we only consider one of the three possible configurations for the final triplet state. The Coulomb interaction between the incident electron and the two electrons in the molecule is

$$V_{\text{e-e}} = \frac{e^2}{4\pi\epsilon_0} \left(\frac{1}{|r_1 - r_3|} + \frac{1}{|r_2 - r_3|} \right).$$

With these notations defined, the transition amplitude

for this spin-exchange inelastic scattering can be formally written as

$$\begin{aligned}T_{i \rightarrow f} &= \langle \Psi_f | V_{\text{e-e}} | \Psi_i \rangle \sim \int \varphi_{\text{H}}^*(r_1)\varphi_{\text{H}}(r_1)dr_1 \times \frac{e^2}{8\pi\epsilon_0} \int \phi_{k_f}^*(r_2)\varphi_{\text{L}}^*(r_3) \frac{1}{|r_2 - r_3|} \varphi_{\text{H}}(r_2)\phi_{k_i}(r_3)dr_2dr_3 \\ &= \frac{e^2}{4\pi\epsilon_0} \int \phi_{k_f}^*(r_2)\varphi_{\text{L}}^*(r_3) \frac{1}{|r_2 - r_3|} \varphi_{\text{H}}(r_2)\phi_{k_i}(r_3)dr_2dr_3.\end{aligned}$$

We can see that the IES-induced excitation from S₀ to T₁ indeed requires a substantial overlap between the wavefunctions of the incident electron and the molecule, which ensures that the above integral is nonzero. Thus, this form of “exchange” interaction becomes effective only when the electron is very close to the electron cloud of the target molecule so that they can “exchange”. Typically, this condition is met when the tip is positioned directly above the molecule, enabling tunneling electrons from the tip to traverse through the molecule. In fact, such a spin-exchange process via IES imposes no requirement on the strength of intramolecular spin-orbit coupling. As demonstrated previously for molecules lacking significant spin-orbit coupling such as H₂Pc and pentacene, their triplet transitions have been experimentally observed in inelastic electron tunneling spectroscopies within metal–insulator–metal tunnel junctions [see, e.g., *Phys. Rev. B* 18, 4241 (1978) and *Phys. Rev. B* 15, 750 (1977)].

Fig. R3. Schematic illustrating the spin-exchange inelastic scattering between an incident electron and the molecule. The molecule is excited from the ground singlet state (S_0) to the triplet state (T_1) after the scattering process. Different colors and labels are used for the incident and scattered electrons to show that the incident electron exchanges with the electron in the molecule.

In the revised manuscript, we have added a phrase “(see **Supplementary Section S5 for more discussion**)” in the main text on page 7, line 158 to provide qualitative discussion on the mechanism of the direct excitation of spin triplets via IES. We have also added detailed discussion on the mechanism of the direct excitation of spin triplets via the spin-exchange IES mechanism in Supplementary Section S5 (pages 13–14, lines 280–316).

Comment 3: *Based on the previous question, I wonder that if the key intermediate relay state (such as the charged state and the T_1 state) as well as the alignment between molecular orbital and Fermi energy level of electrodes could be detected or characterized experimentally? I admit that the characterization to the key intermediate in the multi-step electron transport is challenging, but would provide more evidences to support the electroluminescence diagram.*

Response: Characterizing key intermediate relay states, such as charged states and the T_1 state, as well as the alignment between molecular orbitals and the Fermi energy levels of the electrodes, is indeed a challenging endeavor but holds great potential to provide additional evidence supporting the electroluminescence diagram. In cases where the bias drop in the NaCl layers can be considered negligible, the energy level alignment between the frontier molecular orbitals and the Fermi level of the substrate can be characterized through differential conductance measurements. For instance, the presence of two peaks in the inset of main-text Fig. 1(c) corresponds to the energy difference between the HOMO (LUMO) and the Fermi level of the substrate. When the HOMO (or LUMO) state becomes energetically accessible, the molecule can transiently become charged, and these peaks signify the existence of charged states. Additionally, the presence of charged states can also be detected through electroluminescence spectra, as demonstrated by the cationic emission in Supplementary Fig. S9 for $H_2Pc/NaCl/Au(111)$ [as also reported by Rai *et al.* (*Nano Lett.* 20, 7600–7605 (2020))].

In contrast, the characterization of the triplet relay state is much more challenging. In our model, its existence is mainly based on the agreement between the experimentally detected onset voltage of single-molecule UCEL and the energy of the T_1 triplet state. Nevertheless, weak phosphorescence from the triplet state was observed for the PtPc/NaCl/Ag(100) system, as shown in Fig. R2 in the response to Comment 2 above, but not in the PtPc/NaCl/Au(111) and H₂Pc/NaCl/Au(111) systems, probably due to either larger optical loss of the gold substrate or very weak spin-orbit coupling for H₂Pc. On the other hand, the characterization of the existence of the spin-triplet state experimentally via differential conductance is also highly challenging at the single-molecule level, probably due to very weak signals and resultant poor signal-to-noise ratio. According to a review article [K. W. Hipps and U. Mazur, *J. Phys. Chem.* 97, 7803 (1993)], the IES induced conductance increment in the dI/dV spectrum of the present experiment is estimated to be on the order of a few percent over the value before the IES channel is switched on. As shown in main-text Figs. 1(c) and 4(a), these values are too low to be resolved at the stable tunneling conditions adopted here, with the tunneling current on the order of picoamperes for the single molecules on the NaCl-covered Au(111). Consequently, despite our hard efforts, no inelastic features were detected in the dI/dV spectra, probably due to poor signal-to-noise ratios.

In the revised manuscript, we have broadened the discussion on the difficulties in characterizing key intermediate relay states in UCEL in Supplementary Section S2 (pages 4–5, lines 57–82).

Comment 4: Considering the Ag tip and metal substrate at the experiment, how does the author exclude plasmon-assisted multi-step inelastic scattering scheme (Nat. Photon. 2010, 4, 50–54)? The higher vibrational states may be populated and then enable the upconversion electroluminescence.

Response: The plasmon-assisted multi-step inelastic scattering scheme is indeed a plausible mechanism for upconversion electroluminescence. However, this mechanism is likely much less efficient compared to the triplet-assisted upconversion via the CI+CI mechanism. As we described previously in Ref. [10], the key to the highly-efficient UCEL involves an intermediate relay state with a considerably long lifetime. In our experiments, the typical tunneling currents are smaller than 100 pA, resulting in an average time interval of at least 1 ns between successive tunneling electrons. Consequently, certain relay states with much shorter lifetimes can be reasonably excluded. These include surface plasmons with a lifetime of tens of femtoseconds and molecular vibrational states with a typical lifetime on the order of picoseconds. Both of these states possess lifetimes that are too short to efficiently facilitate the multi-electron UCEL phenomenon. Therefore, we believe that both plasmon-assisted and vibration-assisted upconversion mechanisms can be safely ruled out as the primary mechanism for the anomalously bright UCEL phenomenon here.

In the revised manuscript, we have added a sentence to address this issue (page 2, lines 47–49): “**Plasmon-assisted and vibration-assisted upconversion mechanisms can**

be safely ruled out as their lifetimes are too short to serve as relay states¹⁰”, by referring to previous reasoning given in Ref. [10].

Comment 5: *In the SI, with populated LUMO, the original molecular HOMO can shift upwards owing to the mutual Coulomb interaction. Therein the ΔE_{HOMO} was calculated as $E_{\text{Coul}} + E_{\text{ex}} = \sim 1.5$ eV. This a relatively large shift of the molecular orbital (refer to *J. Am. Chem. Soc.* 2018, 140, 70–73, ~ 0.2 eV of HOMO shift by Coulomb interaction). The detailed mechanism should further be discussed.*

Response: We thank the reviewer for this insightful comment. It is known that the molecular energy levels can undergo significant modifications when the molecule is located within a metal nanogap, as discussed in the literature by K. Moth-Poulsen and T. Bjørnholm [see, e.g., *Nat. Nanotechnol.*, 4, 551 (2009)]. The charging energy or Coulomb repulsion energy of an isolated molecule in the gas phase typically amounts to a few electronvolts. However, within a metal nanogap, this value can be reduced to below 1 eV due to several effects, including the influence of the image potential generated by the substrate and the electronic hybridization with the metal electrodes.

In our previous work (Ref. [10]), we have conducted DFT calculations to estimate the fundamental gap (also known as the transport gap) of an isolated H₂Pc to be approximately ~ 4.1 eV. When the molecule is placed close to the metal substrate, its fundamental gap is greatly reduced due to the image potential effect. Experimentally, the fundamental gap of H₂Pc on 3ML-NaCl/Au(111) is reduced to ~ 2.7 eV, estimated from the onset voltages of the HOMO and LUMO peaks in the dI/dV data (disregarding the bias drop within the NaCl spacer). Consequently, the intermolecular Coulomb repulsion energy is estimated to be reduced to ~ 0.9 eV for a singlet exciton energy at ~ 1.8 eV. According to the energy diagram shown in Supplementary Fig. S4 and discussion there, the energy of the hole to inject into the new “HOMO” (or extracting the electron from the new “HOMO”) to form a T₁ triplet state has to include an additional term of ~ 0.6 eV associated with the electron exchange energy. As a result, the new HOMO can be raised up by a total amount of ~ 1.5 eV. The pure intramolecular Coulomb interaction energy of ~ 0.9 eV (referring to the formation of a S₁ state) is indeed larger than that reported in the paper by Zhou *et al.* (*J. Am. Chem. Soc.* 2018, 140, 70–73). But such a difference in the estimated intramolecular Coulomb repulsion energy could potentially be attributed to the distinct environmental conditions experienced by the molecule. In our experimental setup, the vacuum gap on the tip side and the NaCl spacer on the substrate side effectively decouple the molecule from the metal electrodes. In contrast, in the paper by Zhou *et al.* (*J. Am. Chem. Soc.* 2018, 140, 70–73), PTCDI molecules are attached to the Au-electrode via Au–amine bonds, which may significantly reduce the intramolecular Coulomb repulsion energy.

In the revised manuscript, we have broadened our discussion on the intramolecular Coulomb interaction energy upon charging within a metal nanogap, specifically, in Supplementary Section S4 (page 9–10, lines 201–206): “**Note that the molecular energy levels can undergo significant modifications when the molecule is located**

within a metal nanogap (see, e.g., Refs. 14,15). The charging energy or Coulomb repulsion energy of an isolated molecule in the gas phase typically amounts to a few electronvolts. However, within a metal nanogap, this value can be reduced to below 1 eV due to several effects, including the influence of the image potential generated by the substrate and the hybridization with the metal electrodes.” Two related references (Refs. [14] and [15]) have also been added there.

Comment 6: In Fig. S6, the top panel of IE+CI (via D_0^-), Step 2 shows the electron transport from tip to HOMO. Considering the theory of Marcus electron transfer, the larger energy difference between the electron donor and acceptor would cause a relatively slow rate of electron transfer. In other word, the energy level mismatch to some extent (although it does not violate the energy conservation). Is this electroluminescence diagram more efficient than multi-step IES?

Response: We thank the reviewer for this insightful comment. In response, we would like to emphasize that the carrier injection process involves many-body Coulomb interactions, making the single-electron picture less appropriate for a complete understanding. While energy conservation is essential, the primary factors that influence the rate of this process are the interaction strength between the incident electron and the molecule, as well as the spatial overlap of the various wavefunctions involved in the transition amplitude. Therefore, the energy difference between the Fermi level of the tip and the molecular HOMO might not be a major limiting factor for this process.

As regards the excitation efficiency, the IES+CI mechanism (via D_0^-) shown in the top panel of Supplementary Fig. S6 (which is Supplementary Fig. S8 in the revised version) is believed to be much more efficient than the multi-step IES mechanism. As we discussed in our previous paper (Ref. [10]), IES is an inefficient excitation mechanism due to the very short electron–molecule collision time, compared with the CI mechanism where the electron is trapped in the molecule for a much longer time. Based on a comparison between theoretical simulations and experimental observations for the $H_2Pc/2ML-NaCl/Ag(100)$ system, the IES excitation efficiency for the S_0-S_1 process is about 10% of the latter process via CI. Further simulations indicate that the upconversion efficiency via IES+CI is about 16 times more efficient than IES+IES taking into account different excitation efficiencies to the triplet state. We note that, in Figs. S8, S10 and S12 of the revised Supplementary Information, we only plot the dominant excitation mechanism at each region, while mechanisms that are less efficient are not shown.

In the revised manuscript, we have added a sentence in Supplementary Section S7.1.1 to note that the IES+CI mechanism is much more efficient than multi-step IES mechanism (page 18, lines 414–416): **“For instance, in the regions labeled as “IES+CI”, although both the IES+CI and the IES+IES mechanisms coexist, the latter is not shown due to its much weaker efficiency⁷”**, with Ref. [7] (*Phys. Rev. Lett.* 122, 177401 (2019)) cited for further information.

Comment 7: *A minor suggestion is the caption of the arrows. There are many arrows in Figures, including electron transfer (solid black), the energy transfer (pink), the excitation (dash black) and photon emission (red), IES (green).... The corresponding illustration of these arrows should be provided in the captions.*

Response: We thank the reviewer for the careful reviewing and constructive suggestions. In the figure captions of the revised manuscript (Fig. 2 in the main text and Fig. S8 in the Supplementary Information), we have provided the description for the arrows used in the figures. Specifically, solid black arrows stand for carrier injections (CI), dashed green arrows for inelastic electron scattering (IES), red arrows for photon emission, vertical dashed lines to illustrate level shifting due to charging/discharging, and pink wavy lines to connect processes that occur simultaneously.

Responses to Reviewer 2:

General Comment: *The authors report on an anomalously strong upconversion photon intensities for H2Pc molecules on 2 and 3 ML NaCl/Au(111). The paper is well written and the proposed model for the mechanism of upconversion is sound. The authors speak of a two orders of magnitude stronger upconversion efficiency than in previous studies which is even stronger than electroluminescence in the normal bias region.*

I cannot recommend publication in the present form, as many of the claims are not substantiated by numbers and only a qualitative discussion is given. Many obvious questions remain unanswered, published literature is not considered and probably mayor points are hidden is some small remarks. I list my comments and questions below in more detail. Nevertheless, if the authors can resolve these issues, the paper might be accepted in Nature Communication.

Response: We thank the reviewer for his/her careful review and constructive suggestions. In the following, we provide a point-by-point response to his/her comments.

Comment 1: *Novelty and correctness of mechanism*

The mechanism of upconversion via triplet formation is not new and has been discussed in the literature (see PRL 122, 177401 by the group of the author). In this respect, the manuscript has nothing new to offer. Claims of novelty in the manuscript regarding the mechanism are not justified (page 7 line 158ff). This takes away one of the mayor claims of the paper.

The news is essentially related to the high efficiency of the upconversion. This requires quantitative comparison (see below). A purely qualitative comparison with the emission in the normal region can result in a relatively bright upconversion, if the emission in the normal region is abnormally low. When the bias exceeds the photon energy, the two step mechanism should not be stopped. It should be operative for all high bias voltages, as the tunneling electrons have a broad energy distribution between the two Fermi levels of the electrodes. The presented model does not account for a drop of light emission at higher biases and the question why the intensity decreases for high bias is not explained in the manuscript. However, an interesting side note (negative differential conductivity on page 13) may explain this. It seems that at high bias, the transport channel is cut (total current drops with bias voltage, i.e. negative differential conductance) and with it the light emission, e.g. by simply charging of the molecule. This could explain a very inefficient emission under normal condition that could be mistaken as a very efficient upconversion.

Response: We thank the reviewer again for the careful review. We are also grateful for his/her insightful comments regarding experimental phenomena and mechanisms behind. In the following, we provide responses to all his/her comments in detail.

1.1 Novelty of our work

Although the present paper appears a follow-up to Ref. [10] (*Phys. Rev. Lett.* 122, 177401 (2019)) on a face level, the regimes of focus are fundamentally and distinctly different, with the current work elevating the UCEL phenomenon to new heights in terms of absolute efficiency and working principle. The major novelties of the present work include: (1) Experimentally, we realize for the first time anomalously bright single-molecule UCEL via controlled engineering of the energy-level alignment, with emission efficiencies improved by more than one order of magnitude over previous studies, even stronger than normal-bias electroluminescence (to be justified below in the responses to Comment 2 below). (2) Theoretically, we discover a new and highly efficient UCEL mechanism, which, although still involving a spin triplet state as the relay state, operates in an incredible two-electron excitation mechanism that only involves pure carrier injection steps, without invoking the inefficient IES process proposed in the previous work (to be detailed in the next paragraph). (3) Inspired by this new mechanism, we have further developed a delicate model to construct the electroluminescence diagrams that provide a vivid and straightforward view on the excitation processes of single-molecule electroluminescence encompassing the UCEL.

To be more specific, although the role of spin-triplet state in single-molecule UCEL seems obvious, the microscopic mechanism of how this happens is not trivial at all, as it requires careful analysis and deep thinking. For instance, quite a few mechanisms for UCEL involving triplet states have been proposed in previous literatures, including triplet-triplet annihilation (TTA), thermally activated delayed fluorescence (TADF), IES+IES, IES+CI, etc. Although it was previously found that single-molecule UCEL can be realized via the IES+CI mechanism, it is still a very inefficient mechanism (see Ref. [10]). Therefore, it is non-trivial to experimentally verify the existence or accessibility of highly efficient UCEL in a single molecule and to theoretically interpret the mechanism behind. As a matter of fact, the microscopic mechanism for anomalously bright upconversion via the CI+CI mechanism proposed in the present work is very complicated and not straightforward at all, as it involves multiple carrier injection steps between different intermediate charge states, i.e., neutral ground-state singlet $S_0 \rightarrow$ anionic $D_0^- \rightarrow$ triplet $T_1 \rightarrow$ cationic $D_0^+ \rightarrow$ neutral excited-state singlet S_1 (see main-text Fig. 2(b) for schematic illustration). The CI+CI mechanism is fundamentally different from the IES+CI mechanism previously reported. Moreover, the excitations from S_0 to T_1 and from T_1 to S_1 require critical energy level alignments at the interfaces [see the red triangular area in main-text Fig. 3(b)]. That is probably why such anomalously bright UCEL has not been reported before to the best of our knowledge and the underlying microscopic mechanism is so difficult to discover.

In the revised manuscript, we have added a few sentences in the Conclusion to highlight the major novelty of the present work (page 14, lines 322–329): **“For the anomalously**

bright UCEL to occur, the excitations from S_0 to T_1 and from T_1 to S_1 via the CI+CI mechanism require critical energy level alignments at the interfaces, which enables the amazing transitions between different charge states, namely, neutral ground-state singlet $S_0 \rightarrow$ anionic $D_0^- \rightarrow$ triplet $T_1 \rightarrow$ cationic $D_0^+ \rightarrow$ neutral excited-state singlet S_1 . This is probably why such anomalously bright single-molecule UCEL has not been reported before and the underlying microscopic mechanism is so difficult to discover”.

1.2 Quantitative comparisons regarding the efficiency of single-molecule UCEL

When we state anomalously bright single-molecule UCEL in the present work, we refer to the following two aspects. First, its emission intensity is much stronger than those reported in previous single-molecule studies in the upconversion regime. Second, its emission intensity is even stronger than that in the normal bias (i.e., voltages that surpass the energy of the molecular singlet exciton). We agree with the reviewer that to justify such a claim, quantitative comparisons with previous works (including works from other groups) are necessary. These will be provided in detail in the responses to Comment 2 below.

1.3 Drop of light emission intensities at higher biases

As discussed in Supplementary Fig. S7, the decrease of the electroluminescence intensity at higher biases when $V_b > 1.7$ V is likely due to the increased tip–molecule distance. Although the CI+CI mechanism is still operating, the increase in the tip–molecule distance will decrease the strength of the local plasmonic field and reduce the plasmon–exciton coupling strength. As a result, the electroluminescence efficiency will be decreased, yielding a drop of STML intensities at higher biases.

On the other hand, the negative differential conductance (NDC) is indeed observed for the $H_2Pc/2ML-NaCl/Au(111)$ system, but it occurs only at high bias voltages above ~ 2.3 V. Thus, the occurrence of NDC cannot explain the drop of light emission starting at lower bias voltages (e.g., 1.8–2.2 V) and therefore should not be the mechanism for the decrease of STML intensities at higher biases.

Comment 2: Quantitative data

*The authors do not provide quantitative data on the phonon spectra. In most of the figures, they show spectra as function of photon energy in units of counts/nC. Correct units for spectral data would be e.g. counts/(charge*photon energy). A spectrum reflects the photon spectral density as function of photon energy. Only the integral over the photon energy has the dimensions of counts/nC. Depending on the bin size of the spectra, the plotted numbers can correspond to high or low photon flux. The strong claim of the authors call for a comparison of their intensities quantitatively with results of other systems and other groups in meaningful physical units. Without this direct comparison, the claim of unusually high efficiencies of upconversion cannot be held. I also urge the authors not to normalize the photon intensities but to give the real data*

(Fig. 1c, Fig. 4b etc.).

Response: We thank the reviewer for pointing out the correct unit for STML spectra. We have now replotted all the photon emission spectra using this unit. In Fig. R4, we compare the single-molecule STML spectra presented in this work with those reported from two other groups that are available for direct comparison using the same correct units. As shown in Fig. R4, the anomalously bright UCEL in our work is over ten times stronger than the UCEL reported by Doppagne *et al.* from the ZnPc/NaCl/Au(111) system [Fig. R4(b)] or by Rai *et al.* from the H₂Pc/3ML-NaCl/Au(111) system (see more details in the response to Comment 3 below) [Fig. R4(c)]. In addition, the STML intensity at normal biases (i.e., biases above the molecular optical gap) in our work is found to be comparable to the STML intensities for several different molecules at normal biases reported by other groups [see e.g., Fig. R4(d) and (e)], although the excitation mechanisms might be different.

On the other hand, we believe that a better and more reliable method to quantify the effectiveness of the CI+CI UCEL mechanism is to compare it with other known UCEL mechanisms (e.g., IES+CI) using exactly the same molecule and the same tip, as we have discussed in the main text. As quantified in main-text Figs. 1(b) and (c) (see also the schematic in Fig. 2), when the bias voltage is increased from 1.5 V to 1.7 V, the dominant UCEL mechanism changes from IES+CI to CI+CI, leading to a boost of the UCEL intensity by over two orders of magnitude. Since other experimental conditions are almost exactly the same, the increase in the photon intensity directly reflects the huge increase in molecular excitation efficiencies. Thus, this comparison probably gives a more reliable estimation of the effectiveness of the new CI+CI UCEL mechanism as compared to the IES+CI mechanism previously reported.

Fig. R4. Single-molecule STML spectra from representative systems. (a) STML spectra acquired from $\text{H}_2\text{Pc}/3\text{ML-NaCl}/\text{Au}(111)$ at three different biases (reported in this work). (b) STML spectra acquired from $\text{ZnPc}/3\text{ML-NaCl}/\text{Au}(111)$ (adapted from the reference by Doppagne *et al.*: *Science* 361, 251–255 (2018)). (c)–(d) Bias-dependent STML spectra acquired from a similar system as (a) (adapted from the reference by Rai *et al.*: *Nano Lett.* 20, 7600–7605 (2020)). (e) STML spectra acquired from three different molecules adsorbed on 3ML-NaCl/Ag(111) (adapted from the reference by Cao *et al.*: *Nat. Chem.* 13, 766 (2021)).

We thank the reviewer for the suggestions to provide data with STML intensities that are detected, rather than normalized ones. Figure 1(c) in the previous version is now redrawn following this suggestion, as shown in Fig. R5(a). Nevertheless, because of the relative large differences in emission intensities, in order to highlight the fine tuning of the energy-level alignment through the control of spacer layer thickness, we prefer to use normalized photon intensities in main-text Fig. 4(b). Such a tuning feature is not so evident if plotted in absolute integrated photon intensities, as shown in Fig. R5(b)–(d). The information on the absolute photon emission intensities of these systems are available from STML spectra in Supplementary Fig. S13.

Fig. R5. Photon intensities integrated over the S_1 peak for various single-molecule systems. (a) $H_2Pc/3ML-NaCl/Au(111)$, 30 pA, 60 s. (b) $H_2Pc/2ML-NaCl/Au(111)$, 50 pA, 60 s. (c) $PtPc/3ML-NaCl/Au(111)$, 100 pA, 10 s. (d) $PtPc/4ML-NaCl/Au(111)$, 10 pA, 30 s.

In the revised manuscript, following the reviewer’s suggestions, we have now replotted all STML spectra using the correct unit “kcts $nC^{-1} meV^{-1}$ ”. We have also added a sentence to note on the comparisons about the emission efficiency in the main text (page 4, lines 105–106): “(Note that the UCEL intensity observed here at $V_b = 1.7 V$ is also much stronger than the works reported by other groups^{12,13}.)”, with two references from other groups also cited there. Based on the comparisons shown in Fig. R4, we have modified the descriptions about the emission efficiency improvement by changing “about two orders of magnitude” to “more than one order of magnitude” in the Abstract (page 1, line 24) and Introduction (page 3, line 65), with two more references from other groups also cited there in the Introduction (Refs. [12] and [13]). Figure 1(c) in the main text is now also redrawn with the y-axis showing the detected rather than normalized integrated photon intensity. The STML spectra for $PtPc/NaCl/Au(111)$ are also replotted using the correct unit in Supplementary Fig. S13.

Comment 3: Comparison with published data of the very same system

It remains totally unclear, why the authors do not compare their data with already published results on the very same system of $H_2Pc/NaCl/Au(111)$ (Nano Lett. 20, 7600 (2020)). This work also includes results of upconversion with spectral densities of about 400 counts/(nC meV) and in the “normal” emission of about 2000 counts/(nC meV), i.e. much higher than the upconversion. Obviously, this report is at odds with this work and this discrepancy needs a detailed discussion. It remains completely unclear, why the two works show such vastly different efficiencies although they are on the very same

material system.

Response: We thank the reviewer for bringing up this interesting issue. This reference was cited in our previous version of the manuscript in the Supplementary Information as Ref. [18]. Indeed, for the “same” system, the STML phenomena presented in our present work are quite different from those reported by Rai *et al.* [*Nano Lett.* 20, 7600 (2020)]. By comparing the experimental data, we notice that the HOMO positions in the dI/dV data are quite different in these two works (see Fig. R6), which may account for the distinct STML phenomena. We would like to note that the dI/dV features in our present work are consistent with those reported by Imai-Imada *et al.* [*Phys. Rev. B* 98, 201403(R) (2018)] for $H_2Pc/NaCl/Au(111)$. In our dI/dV data, the HOMO is peaked at around -1.3 V with an onset at about -1.1 V, while in the *Nano Lett.* paper, the HOMO peak is centered at around -2.3 V. Such a large difference in the HOMO position is probably responsible for the distinct STML phenomena reported, in particular for the absence of the anomalously bright UCEL phenomenon in the *Nano Lett.* work. Specifically, as shown in main-text Fig. 2(b), for the CI+CI UCEL mechanism to operate at positive biases, the T_1 triplet state must lie above the Fermi level of the substrate for carrier injection to occur via step 3. While this requirement for the energy level alignment is satisfied in our present system, it breaks down in the case of the *Nano Lett.* work where the HOMO lies deep below the Fermi level of the substrate. Nevertheless, it is still not clear why the HOMO position is quite different for the “same” system. Perhaps, local perturbations due to defects might be possible reasons for such differences in the HOMO positions, as discussed in the PhD thesis of Grewal for the $PtPc/NaCl/Au(111)$ system (see pages 49–51, <https://www.fkf.mpg.de/7828596/dok128-Thesis-Grewal-2022-EPFL.pdf>).

Fig. R6. Differential conductance data measured on the $H_2Pc/NaCl/Au(111)$ system. (a) dI/dV measured on $H_2Pc/3ML-NaCl/Au(111)$ in this work, (b) dI/dV measured on $H_2Pc/NaCl/Au(111)$ (adopted from the reference by Imai-Imada *et al.*: *Phys. Rev. B* 98, 201403(R) (2018)), (c) dI/dV measured on $H_2Pc/3ML-NaCl/Au(111)$ (adopted from the reference by Rai *et al.*: *Nano Lett.* 20, 7600 (2020)).

In the revised manuscript, we have added the *Nano Lett.* paper as Ref. [13] in the

Introduction of the main text when introducing previous works on single-molecule UCEL. We have also added one sentence in the main text to note on previous STML reports for the “same” systems and possible reasons behind (page 12, lines 278–282): **“We would like to note that different STML phenomena were reported for the “same” H₂Pc/3ML-NaCl/Au(111) system by Rai *et al.*¹³, probably due to different adsorption configurations of H₂Pc molecules on NaCl and resultant different energy level alignments at the molecular interface (see Supplementary Section S7.1.3 for details)”**, with the detailed discussion given in the new subsection in the Supplementary Information (Section S7.1.3).

Comment 4: *Tip condition and negative differential conductance*

The $z(V)$ curve in the supplementary figure S4 looks strange. It seems that the tip retracts by about 500pm when ramping the voltage. This is a very large change if keeping in mind, that 100pm change of tip-sample distance changes the current by about one order of magnitude in STM with materials of typical work functions. Can the authors exclude that the tip is partly insulating or it crashes into the sample at low bias? The z -position strongly reacts on the bias in the region of interest between 1.6 and 1.8 V. This corresponds to the onset of the LUMO which in turn can drive the triplet formation (mechanism II), but up to 2.4 V the z -position only increases. The mentioned negative differential conduction must lead to a decrease of current with bias (this is the definition of negative differential conductance) and the z -position under feedback needs to approach. Why is this not observed? The two measurements in this manuscript contradict each other.

Response: We thank the reviewer for the careful review. The observed tip retraction of ~470 pm in Supplementary Fig. S7(a) is indeed quite large, which reflects a big change in the electron tunneling probability up to several orders of magnitude when molecular LUMO states are available, as shown in the $I(V)$ curve of Fig. R7(a) acquired at the constant height mode. In addition, the apparent height for the H₂Pc molecule at $V_b = 2.5$ V shows a consistent value of about 600 pm (Fig. R7(b)) when molecular LUMO states make contribution to the electron tunneling, which also agrees with the apparent height reported for the “same” system by Rai *et al.* (Fig. R7(c)) [*Nano Lett.* 20, 7600 (2020)]. Therefore, the tip status during these measurements is believed to be operated at normal conditions, conducting and not crashing into the sample.

Regarding the issue on negative differential conductance (NDC), as shown in main-text Fig. 4(a), NDC is indeed observed in the H₂Pc/2ML-NaCl/Au(111) system when the bias voltage is above ~2.4 V, and we do observe a decrease in the tip height there in the $z(V)$ curve when $V_b > 2.4$ V (see Fig. R7(d)), as the reviewer suggested. Nevertheless, the NDC phenomenon was not observed in the dI/dV curve for other three systems including H₂Pc/3ML-NaCl/Au(111) up to 2.4 V, that is probably why the z -position keeps increasing with bias voltages in Supplementary Fig. S7(a). Therefore, the two measurements are consistent with each other.

Fig. R7 (a) $I(V)$ curve acquired at the constant height mode, set point: -2.5 V, 30 pA. (b) STM topograph (2.5 V, 2 pA) showing the apparent height of H_2Pc on $3ML-NaCl/Au(111)$ in the present work. (c) STM topograph (-2.5 V, 2 pA) showing the apparent height of H_2Pc on $3ML-NaCl/Au(111)$ in the work by Rai *et al.* [*Nano Lett.* 20, 7600 (2020)]. (d) Vertical tip retraction (Δz) as a function of the bias voltage ($V_b = 1.0$ – 2.6 V) measured at a constant current of 10 pA above a single H_2Pc molecule on $2ML-NaCl/Au(111)$. Note that the appearance of the small pit above 2.1 V is likely to correlate with the hydrogen tautomerization event within a single H_2Pc molecule.

In the revised manuscript, we have added a sentence in the Supplementary Information (page 18, lines 396–398) to note on the large change in the z -position: “(The observed tip retraction of ~ 470 pm in Fig. S7(a) is quite large, which reflects a big change in the electron tunneling probability up to several orders of magnitude when molecular LUMO states are available.)”

Comment 5: Comparison with negative bias voltage

The authors also supply data on the photon emission in negative bias voltage. A very bright Q_x line is reported, but Figure S7 does not give quantitative information on the intensities. How do the intensities compare between positive and negative bias, quantitatively? Does this comparison support that upconversion is particularly efficient at positive bias voltages or does it suggest a particularly low efficiency in the normal region? How is the upconversion explained at negative bias voltages? Is that in agreement with LUMO transitions as well? If not, how sure can we be that the LUMO involvement at positive bias is correct when upconversion is also seen at negative bias?

Response: Following the reviewer’s suggestion, Supplementary Fig. S7 in the previous version (now Supplementary Fig. S9 in the revised version) has been replotted with quantitative emission intensities expressed in the correct unit. Quantitative comparisons on the emission intensities between positive and negative biases are also provided in the similarly replotted main-text Fig. 1(b) for $H_2Pc/3ML-NaCl/Au(111)$. As one can clearly see there, although the STML intensity for the Q_x peak in the normal bias region (e.g., at $V_b = 2.0$ V) is weaker than the UCEL intensity in the upconversion region at $V_b = 1.7$ V, it is still on the same order of magnitude and is much stronger than the STML

intensities at the negative bias voltages. These data support that the upconversion is particularly efficient at positive bias voltages (over 1.6–1.8 V).

The excitation mechanism for the less efficient UCEL at negative bias voltages is discussed in detail in Supplementary Section S7.1.2 (lower sub-panel in Fig. S10(b)). In short, the excitation mechanism is believed to also invoke a triplet-state-mediated multi-electron process, but with the involvement of a transient cationic excited state D_1^+ , as detailed in the broadened discussion in Supplementary Section S7.1.2 (see our actions made to the Supplementary Section S7.1.2 below).

The UCEL mechanisms for $H_2Pc/NaCl/Au(111)$ at positive and negative biases are evidently different in terms of the first-step carrier injection, but both involve the generation of the S_1 state and subsequent LUMO–HOMO radiative transitions. Specifically, at positive biases, the first carrier injection step occurs via electron injection into the LUMO, whereas at negative biases, the first carrier injection step occurs via hole injection into the HOMO. These carrier injection processes are evidenced by the LUMO and HOMO peaks in the dI/dV curves (see, e.g., main-text Fig. 1(c)). These injected carriers can significantly shift the molecular orbitals due to the nontrivial intramolecular Coulomb interactions, which allows the molecule to be excited to S_1 via subsequent carrier injections into various transient intermediate states. Thus, we believe that the UCEL mechanisms we propose for the positive and negative biases are consistent, both involve the generation of the S_1 state that corresponds to an electronic configuration with one electron in the LUMO and one hole in the HOMO, which recombine to produce photons.

In the revised manuscript, we have replotted Supplementary Fig. S7 (now Fig. S9) using the correct unit. Our actions also include adding a broadened discussion on the UCEL mechanism at the negative bias in Supplementary Section S7.1.2 (page 24, lines 494–510): **“Note that the mechanism shown in region ③ of Fig. S10 refers to both the upconversion and normal-bias electroluminescence. For the UCEL region at negative biases, both the IES+IES mechanism in region ① and the CI+ w -IES mechanism in region ② shown in Fig. S10 are very inefficient, yielding negligible S_1 emission when $V_b \leq -1.45$ V. The dominant upconversion mechanism for the UCEL phenomenon from -1.46 V to -1.81 V is believed to associate with another type of CI+CI mechanism that involves a transient excited state of the intermediate cation D_1^+ . Specifically, to reach the S_1 state, the following four steps are required (see region ③ in Fig. S10). First, the electron in the HOMO tunnels to the tip (step 1), leaving behind a transient ground-state cation D_0^+ (in other words, a hole is injected into the HOMO); Second, the D_0^+ state can be transformed to the neutral T_1 state upon an electron injection from the substrate (step 2); Third, the electron in the “HOMO” of the T_1 state transfers to the lower-lying Fermi level of the tip, with its excess energy transferred to the electron in the “LUMO” of the T_1 state, inducing the molecule to the excited-state cation D_1^+ (step 3); Fourth, another electron in the substrate tunnels into the “HOMO” of D_1^+ , changing the molecule to the neutral excited state S_1 (step 4). Nevertheless, presumably limited by the short lifetime of the transient excited state D_1^+ , the**

UCEL intensity at the negative bias is weaker than that at the positive bias.”

Comment 6: Comparison to NaCl/Ag(100)

The data shown in the supplementary material (S9) for the analogous experiments on Ag(100) has many orders of magnitude more efficient light emission in the normal region (S9a shows Q_x intensities that are indeed high and of the order of 10^8 counts/nC). Also this speaks for a strongly inefficient emission in the normal bias regime for the case of Au(111) and against a very efficient upconversion.

Response: We thank the reviewer for his/her sharp observation and careful reviewing. We apologize for a typo made in Supplementary Fig. S9(a) in the previous version of the Supplementary Information. Specifically, the unit in the y-axis was not properly adjusted when editing the figure labels. The label there should be “Photon intensity (counts/nC)” rather than “Photon intensity (10^4 counts/nC)”. In the revised manuscript, we have corrected this typo and replotted the figure using the correct unit (which is now Supplementary Fig. S11(a), see also Fig. R8 below). With this correction, together with the detailed quantitative comparison we made in the response to Comment 2 above, we hope the reviewer is convinced that the UCEL for the H₂Pc/NaCl/Au(111) system is anomalously bright. We thank the reviewer again for pointing out such a typo.

Fig. R8 STML spectra for H₂Pc/3ML-NaCl/Ag(100) at V_b = -1.7, -2.3, -2.5 V (20 pA, 60 s) and +2.5 V (100 pA, 180 s). Spectral curves are offset for clarity.

Comment 7: Citations

Upconversion has been studied by many groups. I find that the citations are too much biased regarding the authors own publications. Also the claim of anomalous upconversion efficiency needs to be tested against the literature especially of competitors and not exclusively to own work.

Response: Following the reviewer’s suggestion, in the revised manuscript, we have added references on single-molecule UCEL from other groups, specifically, Refs. [12] and [13]. In the response to Comment 2 above, we have made quantitative comparisons

not only to our own work, but also with the works by other groups, to justify the statement of the anomalously bright UCEL.

Responses to Reviewer 3:

General Comment: *In this work, efficient upconversion electroluminescence (UCEL) was observed in a single phthalocyanine molecule, and the emission efficiencies can be improved via engineering the energy-level alignments at the molecule–substrate interface. Moreover, a new spin-triplet mediated UCEL mechanism was proposed for producing efficient UCEL. These findings provide deep insights into the microscopic mechanism of single-molecule UCEL and organic electroluminescence in general.*

Overall, the research is interesting and should be accepted for publication in Nature Communication after addressing several questions related.

Response: We thank the reviewer very much for his/her appreciation of our work and positive recommendations. In the following, we provide a point-by-point response to his/her constructive comments.

Comment 1: *Why was the phthalocyanine molecule selected for this research?*

Response: The selection of planar phthalocyanine molecules for this research was based on several compelling factors. Both the free-base and metal phthalocyanine molecules are widely utilized in the STML community due to their well-defined geometry for identification by STM imaging, remarkable stability for adsorption on NaCl spacer layers, rigid structure and resultant comparatively high electroluminescence intensity. Indeed, this type of molecule has consistently proven to be a “star” system for investigating optoelectronic processes at the single-molecule level by STML.

In the Methods section of the revised manuscript (pages 14–15, lines 343–347), we have added two sentences to explain why the phthalocyanine molecules were selected for this research: “(The selection of planar phthalocyanine molecules for this research was based on several compelling factors. Both the free-base and metal phthalocyanine molecules are widely utilized in the STML community due to their well-defined geometry for identification by STM imaging, remarkable stability for adsorption on NaCl spacer layers, rigid structure and resultant comparatively high electroluminescence intensity.)”

Comment 2: *Does the PtPc show more efficient UCEL efficiency?*

Response: In the previous version of the manuscript, the STML spectra from PtPc/NaCl/Au(111) were plotted in arbitrary units in Supplementary Fig. S11 (now Supplementary Fig. S13 in the revised version), not suitable for quantitative comparisons on the emission intensities. In the revised manuscript, we have replotted all STML spectra using the new unit “kcts nC⁻¹ meV⁻¹” to express the STML intensities detected, thus facilitating quantitative comparisons on the STML intensities between

PtPc/3ML-NaCl/Au(111) (see Supplementary Fig. S13(a)) and H₂Pc/3ML-NaCl/Au(111) (see main-text Fig. 1(b)). According to these STML spectral data, the UCEL intensity for PtPc/3ML-NaCl/Au(111) at $V_b = 1.8$ V appears weaker than the UCEL intensity for H₂Pc/3ML-NaCl/Au(111) at $V_b = 1.7$ V. However, it is noteworthy that these spectra were collected in separate experiments using different STM tips. Since the STML intensities from single molecules are sensitive to the plasmonic properties of the tip, absolute emission intensity comparisons can only be made on the same sample using the same tip, which will be examined in future experiments.

In the revised manuscript, we have added a sentence in Supplementary Section S8 to address the issue on the PtPc UCEL efficiency (page 28, lines 583–585): **“By comparing with the STML spectral data in main-text Fig. 1(b), the UCEL intensity for PtPc/3ML-NaCl/Au(111) at $V_b = 1.8$ V appears weaker than the UCEL intensity for H₂Pc/3ML-NaCl/Au(111) at $V_b = 1.7$ V.”**

In closing, we thank all reviewers for their careful review of this work and their constructive suggestions. We hope we have adequately addressed all their concerns, and have also improved the effectiveness of the presentation in conveying the importance, novelty, and validity of the main findings presented in the paper.

REVIEWERS' COMMENTS

Reviewer #1 (Remarks to the Author):

The authors have answered my comments satisfactorily. I recommend its publication.

Reviewer #2 (Remarks to the Author):

The revised manuscript was improved significantly by the authors. They have very convincingly answered the questions of all referees and have modified the manuscript and supplementary material, so that also the reader benefits from the discussion between authors and referees. Most importantly, the revised manuscript now gives spectral intensities allowing a quantitative comparison. A consequence of this, they reduced their claims to an enhancement of up conversion efficiency by more than one order (down from two orders). This softened claim is solid, well documented and fully explained.

I now do not have any open questions, anymore and support publication of this nice and important paper as is.

Reviewer #3 (Remarks to the Author):

In the revised version, all of my comments and concerns were thoroughly addressed by the authors. Also, I noticed that the critical comments raised by other reviewers were also justified and addressed in detail by the authors. So, I strongly recommend the acceptance of this manuscript for publication in Nature Communications.